# The Role of the Respiratory Microbiome in the Pathogenesis of Aspiration Pneumonia: Implications for Diagnosis and Potential Therapeutic Choices

**DOI:** 10.3390/antibiotics12010140

**Published:** 2023-01-10

**Authors:** Natalia G. Vallianou, Alexandros Skourtis, Dimitris Kounatidis, Evangelia Margellou, Fotis Panagopoulos, Eleni Geladari, Angelos Evangelopoulos, Edison Jahaj

**Affiliations:** Department of Internal Medicine, Evangelismos General Hospital, 45-47 Ipsilantou, 10676 Athens, Greece

**Keywords:** lung microbiome, gram negative bacteria, PCR, mNGS

## Abstract

Although the lungs were considered to be sterile until recently, the advent of molecular biology techniques, such as polymerase chain reaction, 16 S rRNA sequencing and metagenomics has led to our expanding knowledge of the lung microbiome. These methods may be particularly useful for the identification of the causative agent(s) in cases of aspiration pneumonia, in which there is usually prior administration of antibiotics. The most common empirical treatment of aspiration pneumonia is the administration of broad-spectrum antibiotics; however, this may result in negative cultures from specimens taken from the respiratory tract. Therefore, in such cases, polymerase chain reaction or metagenomic next-generation sequencing may be life-saving. Moreover, these modern molecular methods may assist with antimicrobial stewardship. Based upon factors such as age, altered mental consciousness and recent hospitalization, there is a shift towards the predominance of aerobes, especially Gram-negative bacteria, over anaerobes in aspiration pneumonia. Thus, the therapeutic choices should be expanded to cover multi-drug resistant Gram-negative bacteria in selected cases of aspiration pneumonia.

## 1. Introduction

The gut microbiome, which comprises the major component of the microbiome in human beings, refers to the >10^14^ bacteria that inhabit the human intestines. This number encompasses 10 times more bacterial cells than the number of human cells and over 100 times the amount of bacterial genomic content, when compared to the human genome [1,2,3]. In 2008, the National Institutes of Health (NIH), introduced the Human Microbiome Project to study the microbiome in various sites of the human body. The lungs were not among the 15 sites sampled in men and the 18 sites sampled in women in this Project, which included 250 healthy volunteers [4]. However, there is now a growing body of evidence that advocates the existence of the lung microbiome. Although the lungs have long been considered to be sterile in healthy humans, this notion has now changed dramatically. The advent of modern molecular techniques has led to the denial of this widely accepted misconception. Polymerase chain reaction (PCR), 16 S rRNA sequencing and metagenomics have revolutionized our understanding of the lungs’ microenvironment among healthy adults. Indeed, in 2010, Hilty et al. were the first to publish a study of the lung microbiome in healthy participants and patients with asthma, which used these techniques. Based on observations of bronchoalveolar lavage (BAL) and brushing specimens, they reported that the lung microbiome was similar, but distinct, to the upper respiratory tract microbiome [5]. Since then, many researchers have contributed to the study of the lung microbiome. It has now been demonstrated that the lung microbiome is less abundant in healthy adults compared to the gut microbiome and the upper respiratory tract microbiome [5,6,7].

Aspiration pneumonia refers to the development of a lung infection due to a specific microorganism(s), which takes place after the aspiration of oropharyngeal or upper gastrointestinal content [8]. It is usually located according to gravity in the superior lower lobe or in the posterior upper lobe segments, if the patient is in the supine position. It should be differentiated from chemical pneumonitis, which refers to the irrigation of the lung parenchyma due to an aspiration event, but not to the inflammation caused by bacteria [8]. It is estimated to account for approximately 15% of all cases of community-acquired pneumonia, whereas the estimation rates for nosocomial-acquired cases vary largely due to the heterogeneity of the populations studied [9]. In addition, it may be difficult to diagnose aspiration pneumonia due to the lack of the availability of non-cultural methods in the clinical setting [5].

In this review, we aimed to describe the relationship between the lung microbiome and the pathogenesis of aspiration pneumonia and to focus on the diagnostic and therapeutic potential of the lung microbiome regarding aspiration pneumonia.

## 2. The Lung Microbiome in Health

### 2.1. The Lungs Are Not Sterile

Although the linear distance from the nares to alveoli is about half a meter, the total internal lung surface area is approximately 30 times larger than that of the skin [10]. In addition, the total internal lung surface is estimated to be almost double that of the gut [11]. In 1896, Thomson and Hewlett reported that a total of up to 14,000 organisms are inhaled per hour [12]. In addition to the huge number of inhaled microorganisms, it is now widely accepted that microaspirations can also occur in healthy adults. In fact, subclinical microaspirations are common events during sleep, even among healthy individuals [8,12,13,14]. During sleep, the body is in the supine position and the laryngeal and coughing reflexes are reduced. Taking these considerations into account, microaspirations during sleep seem to be inevitable even among healthy adults [8].

The observations above are crude estimates of a large lung surface, which could harbor a dynamic lung ecosystem where microbial immigration and elimination factors both play a crucial role. In particular, microbial immigration refers to the inhalation of various microbes from the air as well as microaspirations from the oropharyngeal and the upper respiratory tract microbiome to the lungs [8,12,13,14]. Microbial elimination is driven by the functions of mucociliary clearance, coughing, and host immune defense mechanisms [15,16,17]. In the lungs of healthy adults, bacteria exist in significantly less abundance than in the oropharynx. This may be attributable to the effectiveness of the ciliated epithelium, resulting in an outflow of bacteria via the mucus, coughing and host immune defense mechanisms [15,16,17,18]. Under normal circumstances, there is balance between microbial immigration and elimination, which results in the unique composition of the lung microbiome in each and every healthy individual (Figure 1). The lung microbiome of healthy adults is characterized by increased bacterial diversity, but low abundance when compared with the gut microbiome. Studies have indicated that three main genera exist in the lungs of healthy individuals: *Prevotella*, *Veillonella* and *Streptococcus*. On the contrary, under abnormal circumstances, an imbalance between the immigration and elimination mechanisms may lead to the development of a lung microbiome that is characterized by reduced bacterial diversity and/or the predominance of a small group of taxa or species (Figure 2) [17,18,19,20,21]. There is mounting evidence for the reduced diversity in lung microbiome in various lung diseases such as asthma, pneumonia, chronic obstructive pulmonary disease, cystic fibrosis and lung cancer [21,22]. The diversity of the lung microbiome increases with age when compared to the lung microbiome of children, as is also the case for the gut microbiome. However, for example, in the case of cystic fibrosis, Linnane et al. demonstrated a decrease in the diversity of the lung microbiome with age among patients with cystic fibrosis. In addition, they showed that *Pseudomonas* and *Staphylococcus* were more abundant with age. On the contrary, *Streptococcus*, *Porphyromonas* and *Veillonella* were encountered less frequently with age among patients with cystic fibrosis [23].

### 2.2. The Role of Non-Cultural Methods in Describing the Lung Microbiome

Our understanding of the lung microbiome has expanded lately due to the development of non-cultural methods for the identification of microorganisms. Conventional cultural methods are not able to detect all of the microorganisms that comprise the lung microflora, and they can also be time-consuming. Therefore, the advent of non-cultural molecular techniques for studying the lung microbiome have greatly revolutionized our knowledge of the diversity of the lung microbiome in healthy adults. PCR, 16 S rRNA gene sequencing, and especially whole-generation sequencing and metagenomics, can provide us with information to the species level, and all of these methods have contributed to our defining the composition of the lung microbiome in healthy individuals [24,25]. Shotgun metagenomic sequencing has given us the opportunity to sequence all genetic material within a sample, thereby allowing for a holistic and deeper taxonomic characterization of the lung microbiome to the species or even strain level [24,25].

## 3. The Lung Microbiome in Aspiration Pneumonia

It has been reported that the lung microbiome changes dramatically during disease [26,27,28]. Apart from disruptions in microbial immigration and elimination, other features implicated in the alterations of the lung microbiome during disease may be attributable to differences in local environmental factors along the respiratory tract. Temperature variations, pH, oxygen concentration, nutrient availability as well as host defense parameters, such as host epithelial cell interactions and local inflammatory responses, may result in dysbiosis in the lung microbiome [12,26,27,28]. The term “dysbiosis in the lung microbiome” is used to underline the differences in the composition of the microbial communities in the lower respiratory tract seen in disease as compared to the state of homeostasis between the bacterial communities and the host seen under healthy conditions [26,27,28].

Aspiration pneumonia results mainly from the macroaspiration of oropharyngeal or gastrointestinal content in patients with impaired swallowing or reduced cough reflexes [8]. Risk factors associated with the development of aspiration pneumonia include neurological disorders as well as gastrointestinal diseases and several medications, such as sleeping pills or anesthetics. The risk factors related to aspiration pneumonia are listed in Table 1.

Regarding aspiration pneumonia and the lung microbiome, the phenomenon of “dysbiosis” has been documented in many studies [8,29]. In particular, due to impaired swallowing or/and impaired coughing reflexes, in most cases of aspiration pneumonia there is a swift from anaerobes to Gram-negative bacteria [8,29]. Indeed, the most frequently encountered bacteria in aspiration pneumonia are aerobes, especially Gram-negative rods such as *Escherichia coli*, *Klebsiella pneumoniae* and *Pseudomonas aeruginosa* [8]. These Gram-negative bacteria may be resistant to various antibiotics. More specifically, resistance to Gram-negative bacteria has recently been reported as less than 10% in most European countries [29]. Although these percentages may seem to be somewhat low, they should be interpreted with caution. According to the CDC, antibiotic resistance accounts for more than 25,000 deaths annually in Europe, with the vast majority being attributed to Gram-negative bacteria [30].

The pathogenesis of aspiration pneumonia is considered to be the result of the interplay between the existence of the low diversity, high biomass lung microbiome and the host immune responses [29]. The cornerstone of the pathogenesis of aspiration pneumonia is macroaspiration, which leads to the abundance of one or a few species in the lung microbiome. This abundance interacts with the host’s responses, such as the recruitment of inflammatory cells as well as disturbances in the production of cytokines and chemokines, resulting in a dysregulated local immune response. The phenomenon of the dysregulated host immune response plays a crucial role in the development of aspiration pneumonia [29]. All the above-mentioned contributing factors interact with each other, and lead to the pathogenesis of aspiration pneumonia.

## 4. Diagnosis of Aspiration Pneumonia

### 4.1. Diagnosing Aspiration Pneumonia Is Mainly Clinical

The diagnosis of aspiration pneumonia is usually based on clinical grounds such as the witness of a macroaspiration event, especially among patients with predisposing factors, as has already been mentioned above. In addition to the witness or history of an antecedent macroaspiration event, hypoxemia and crackles detected during chest auscultation also point to the diagnosis of aspiration pneumonia. A chest X-ray showing infiltrates may be compatible with the diagnosis of aspiration pneumonia, especially if the infiltrates are located according to gravity parameters. However, a chest X-ray may be negative at the time of the diagnosis of aspiration pneumonia in up to 30% of the cases confirmed by thoracic computed tomography [8,31].

### 4.2. Microbiology in Diagnosing Aspiration Pneumonia: Conventional versus Modern Molecular Methods

Culture-based techniques using sputum or trachiobronchial aspirates may reveal the presence of Gram-negative bacteria in the vast majority of cases. The clinical specimen is cultured on an agar plate and then left for overnight aerobic incubation at 37 °C. After bacterial culture, and based upon the various biochemical features of the bacteria, identification to the species level is performed, usually with the use of API Systems. Nowadays, identification to the species level may be achieved for highly complex microorganisms by using the more sophisticated matrix-assisted laser desorption and ionization time of flight mass spectrometry (MALDI-TOF MS) [32]. In addition, antibiotic susceptibility testing is usually performed, mainly by using the disk diffusion technique. However, as already mentioned above, culture-based techniques are time-consuming and may not yield any microorganisms, especially when antibiotics have already been administered. Prior antibiotics administration is usually the case among patients with nosocomial aspiration pneumonia, as in the elderly or after an operation where general anesthesia has been utilized. Therefore, conventional culture-based techniques may not be helpful under these circumstances.

The advent of non-cultural sophisticated methods appears to be very promising in this context. Multiplex PCR or 16 S rRNA gene sequencing of various Gram-negative bacteria implicated in nosocomial aspiration pneumonia is increasingly being utilized worldwide. In fact, the Infectious Diseases Society of America (IDSA) and the American Thoracic Society advocate the usage of more accurate and more rapid molecular methods for the detection of the etiologic agent(s) of respiratory tract infections, especially under selected circumstances [33,34,35]. Just recently, Darie et al. suggested that the use of multiplex bacterial PCR in bronchoalveolar lavage among patients admitted with pneumonia and at risk for Gram-negative bacteria has decreased the duration of empirical treatment [36]. Darie et al. concluded that this reduced duration of sometimes inappropriate treatment might have significant implications regarding antibiotic stewardship [36]. Currently, the administration of broad-spectrum antibiotics in cases of aspiration pneumonia is common practice. Antibiotic stewardship requires that if cultured-based techniques are used, de-escalation should be performed after 48–72 h according to the cultured-based results. However, as these conventional techniques may turn negative, the usage of modern non-cultural-based molecular methods should be encouraged under specific circumstances. Other researchers have also recommended the increasing use of molecular techniques for the purposes of antibiotic stewardship [37,38]. For example, it is widely known that carbapenem-resistant *Acinetobacter baumannii* (CRAB) is a major cause of hospital-acquired pneumonia (HAP) and ventilator-associated pneumonia (VAP). Yin et al. documented a high prevalence of *Acinetobacter baumannii* among patients with VAP in China between 2007 and 2016. Notably, most strains of *Acinetobacter baumannii* were demonstrated to be multi-drug resistant or even pan-drug resistant [39]. They concluded that the rational and tailored use of antibiotics could play a pivotal role in containing the spread of CRAB [39]. Additionally, Xiao et al. highlighted the difficulties in distinguishing CRAB as a pathogen or a member of the microflora of the lung microbiome. In particular, as *Acinetobacter baumannii* is a ubiquitous microorganism, it may be frequently isolated from the respiratory tract in patients with tracheal intubation. However, this does not necessary translate into the existence of VAP due to *Acinetobacter baumannii*; rather, it may just indicate a colonization. On the other hand, an accurate diagnosis of CRAB pneumonia is mandatory as it is associated with substantial mortality [40]. However, PCR has some limitations, such as its inability to discriminate between viable and non-viable microorganisms and it may not be able to distinguish between a colonization and an infection. Therefore, to distinguish between CRAB colonization and infection, and thereby to reduce any unnecessary use of antibiotics, Xiao et al. used multi-omics analysis and performed 16 S rRNA amplicon analysis, metagenomics sequencing and whole genome sequencing (WGS) [40]. These techniques, with the advent of specific platforms and bioinformatics, are very helpful in providing us with information regarding CRAB as a colonizer or a pathogen in respiratory tract infections. The authors concluded that patients with CRAB pneumonia were characterized by decreased diversity in the lung microbiome, increased relative abundance of *Acinetobacter* and the increased existence of virulence factors [40].

While antibiotic stewardship is of the outmost importance in terms of preventing the alarming increase in the spread of antimicrobial resistance, there are many other reasons why non-cultural molecular methods should be tried under specific circumstances. In particular, modern molecular methods are fast and precise in diagnosis. In addition, especially in cases of fastidious microorganisms, they provide us with timely results that cannot be obtained with the use of conventional cultural techniques. Furthermore, the prior use of antibiotics, which is common practice, often results in the failure to reveal the etiological agent(s) of aspiration pneumonia with culture-based techniques. For example, Wang et al. used two different metagenomics next-generation sequencing (mNGS) platforms to confirm *Klebsiella pneumoniae* as the causative agent, harboring antibiotic resistance genes *blaSHV-12*, *aac(3)-IIa* and *blaKPC-2*. The identification of *Klebsiella pneumoniae* and its antibiotic resistance genes by mNGS, led to the successful treatment of this immune-compromised patient with culture-negative pneumonia [41]. Therefore, under specific circumstances, especially among immune-compromised patients or in cases of prior use of antibiotics, the use of modern molecular methods seems mandatory in order to establish the causative agent(s) and to identify gene mutations conferring antimicrobial resistance.

In particular, mNGS makes it possible to identify new microorganisms. This holds true for the identification of SARS-CoV-2 on 3rd February 2020 in Wuhan, China, from three distinct cases of severe pneumonia, with the use of RNA-based mNGS [42]. It is notable that the SARS-CoV-2 genome was identified within 5 days whereas SARS-CoV was sequenced during a five month period [43]. Zhu et al. used a combination of the Illumina and the nanopore sequencing systems to identify the genome of SARS-CoV-2 [42].

In the era of climate change, there is always the possibility that novel bacteria will be implicated in cases of aspiration pneumonia, which may disseminate throughout the world in the near future. Additionally, new mutations in already known bacteria that are associated with antimicrobial resistance are still emerging worldwide [44]. RNA-based mNGS provides the opportunity to explore the expression of genes of the lung microbiome; thus, differentiating between aspiration pneumonia and chemical pneumonitis due to aspiration. More specifically, RNA-based mNGS could discriminate an infectious from a non-infectious cause of aspiration pneumonia or chemical pneumonitis, respectively. Therefore, it would be useful in addressing the issue of the administration of antimicrobial chemotherapy, or not, within 48 h or even within six hours, when using the nanopore sequencing technology with rapid turn-around times [44,45]. Thereby, it would spare us the unnecessary overuse and misleading use of antibiotics, thus resulting in reductions in the emergence of new antimicrobial resistance genes.

## 5. Treatment of Aspiration Pneumonia

### 5.1. Considering the Past Available Antimicrobial Agents

Currently, in cases of aspiration pneumonia, there is a shift towards Gram-negative aerobe bacteria instead of anaerobes [8]. However, the etiologic agent(s) in aspiration pneumonia varies significantly according to previous hospitalizations or history of residing in health care facilities, the consumption of broad-spectrum antibiotics within the last 3 months, and the duration of current stay in the hospital. In particular, a current hospitalization of more than 5 days or a recent previous one or the administration of antibiotics during the past 3 months have been associated with the acquisition of multi-drug resistant Gram-negative bacteria, which may be difficult to treat [8]. On the contrary, the absence of the above-mentioned factors is related to aspiration pneumonia caused by the usual flora of the individuals and not to multi-drug resistant strains. Therefore, in order to contain the spread of antibiotic resistance and for more accurate decision making with regard to the use of antibiotics, it is prudent to send a clinical specimen for identification of the causative agent(s) as well as for susceptibility testing to various antibiotics. However, in most cases of aspiration pneumonia, the administration of ampicillin-sulbactam or a fluoroquinolone. such as levofloxacin or moxifloxacin or a third-generation cephalosporin, such as ceftriaxone, is highly recommended. However, if antibiotic resistance is a concern, as in the case of hospital-acquired aspiration pneumonia, the use of piperacillin-tazobactam, levofloxacin, a fourth-generation cephalosporin. such as cefepime or a carbapenem, such as meropenem or imipenem is advocated [8]. Moreover, in their review article in New England Journal of Medicine in 2019, Mandell and Niederman recommended the addition of colistin or an aminoglycoside to the abovementioned regimens in difficult-to-treat aspiration pneumonia [8].

### 5.2. Considering Newer β-lactamase Inhibitors Combinations

It is crucial to identify the multi-drug resistant strains responsible for the difficult-to-treat cases of aspiration pneumonia. Apart from conventional antimicrobial susceptibility testing, newer molecular-based techniques may contribute to the determination of susceptibility, or not, as already mentioned above.

Nowadays, as well as in the past, the initiation of *β*-lactamase inhibitors (BLIs) is used as the main strategy to reinforce the usage of *β*-lactams. After a period of no significant advances in the field of BLIs, a new era has now dawned with regard to BLIs [46]. Newer *β*-lactamase inhibitor combinations, such as ceftazidime–avibactam, aztreonam–avibactam, meropenem–vaborbactam, imipenem–cilastatin–relebactam, ceftolozane–tazobactam and cefepime–enmetazobactam are already being used worldwide or are eagerly anticipated [46,47,48,49,50,51].

#### 5.2.1. Avibactam

Avibactam is a non-*β*-lactam BLI that has activity against class A *β*-lactamases, namely, KPC, TEM, CXT-M, SHV, class C *β*-lactamases, namely, ampC, and some members of class D serine *β*-lactamases such as OXA-48 and OXA-10. However, it has no activity against class B metalloproteases (MBL) such as VIM, NDM, and IMP [46,47,48,49,50,51]. On the contrary, as aztreonam is stable to hydrolysis by class B MBL, the combination of aztreonam–avibactam is a promising candidate in this context [46,47,48,49,50,51].

#### 5.2.2. Vaborbactam

Vaborbactam is a non *β*-lactam BLI that is effective against many serine *β*-lactamases such as KPC [52]. It is a cyclic boronic acid product, which has no antibacterial activity per se. It has been designated to be combined with a carbapenem, such as meropenem, which is known to possess better activity against Gram-negative bacteria than imipenem, with the latter exhibiting better activity against Gram-positive bacteria than the former [53]. Despite the fact that it lacks antibacterial potential, its fixed combination as meropenem–varbobactam has been demonstrated to be very promising in many in vitro studies [54].

#### 5.2.3. Relebactam

Relebactam is a diazabicyclooctane (DBO) *β*-lactamase inhibitor, known for its inhibitory action upon *Klebsiella pneumoniae* carbapenemase-2, KPC-2 *β*-lactamase [55]. KPC-2 is the predominant carbapenemase among carbamapenase-resistant *Enterobacterales* (CRE) in the United States. Moreover, it has already been reported to be widely disseminated across the world [56]. It is resistant to *β*-lactam antibiotics as well as to *β*-lactam-derived BLIs, such as clavulanic acid, sulbactam, and tazobactam as well [57]. Relebactam has been developed in combination with imipenem (in the imipenem–cilastatin–relebactam fixed combination) to overcome the widespread KPC-2 production in CRE. Cilastatin has already been used for many years in combination with imipenem as it works as an inhibitor of renal dehydropeptidase, an enzyme known to hydrolyze imipenem in the kidney, thus resulting in the prevention of the rapid renal metabolism of imipenem. Relebactam exhibits activity against class A and class C *β*-lactamases whilst the fixed combination of imipenem–cilastatin–relebactam seems promising in combating CRE and carbapenemase-resistant *Pseudomonas aeruginosa*, but not carbapenemase-resistant *Acinetobacter baumannii*, (CRAB) [58].

### 5.3. Next-Generation BLIs: Enmetazobactam, Zidebactam, Taniborbactam, Nacubactam and Durlobactam

#### 5.3.1. Enmetazobactam

Enmetazobactam is a new BLI that differs from tazobactam only by the addition of a N-methyl-group, which renders this molecule neutral, thus, facilitating bacterial cell wall entry [59]. In combination with the fourth-generation cephalosporin cefepime, it possesses activity against class A, class C and class D BLIs [60].

#### 5.3.2. Zidebactam

Zidebactam is a novel BLI with a non-*β*-lactam bicycloacyl hydrazide pharmacophore, which exhibits an inhibitory effect on penicillin binding protein 2, PBP2. It has also been combined with cefepime. This combination has good in vitro activity against multidrug-resistant *Pseudomonas aeruginosa* due to its combination with the PBP3 inhibitor cefepime. The effectiveness of the zidebactam *β*-lactam enhancer mechanism is not impacted by the concurrent expression of ESBLs, class C, OXA-48-like, and MBL-carbapenemases, despite the fact that zidebactam is a non-inhibitor of the latter two enzymes [61]. Past in vitro and in vivo studies have established cefepime–zidebactam’s novel mechanism of action, and coverage of MDR *Enterobacterales*, *Pseudomonas* and *Acinetobacter* [62,63,64].

#### 5.3.3. Taniborbactam

Taniborbactam is a bicyclic boronic molecule with *β*-lactamase inhibitory activity against serine-*β*-lactamases (KPC or GES and OXA-48) as well as MBLs including VIM and NDM enzymes, but not against IMP [65,66,67,68]. Taniborbactam has also been developed in combination with cefepime. This combination has promising potential against CRE and carbapenem-resistant *P. aeruginosa* strains, including MBL-producing ones.

#### 5.3.4. Nacubactam

Nacubactam is a novel BLI, which, like relebactam, belongs to DBO type *β*-lactamases and is also structurally related to avibactam. As well as having good activity against serine *β*-lactamases, (class A, class C and some class D *β*-lactamases), it possesses intrinsic antibacterial activity due to its ability to inhibit PBP2 [69]. With these dual mechanisms of action, nacubactam is especially potent in vivo against *AmpC* overproducing and KPC-expressing *Pseudomonas aeruginosa* strains; however, it lacks activity against MBLs [69]. Nevertheless, there are a few studies advocating its activity even against MBLs, when used in combination with a *β*-lactam antibiotic, such as meropenem or aztreonam [70,71,72].

#### 5.3.5. Durlobactam

Durlobactam is the newest BLI and it belongs to the family of DBOs and has activity against serine *β*-lactamases, class A, class C and class D, It has been produced to overcome resistance to carbapenemase-resistant *Acinetobacter baumannii* (CRAB) and for this reason, it has been tried in combination with sulbactam [73]. Sulbactam, a semi-synthetic penicillanic acid with endogenous activity against *Acinetobacter baumannii*, may be effective when used in combination with durlobactam. A sulbactam–durlobactam combination may lead to the restoration of sulbactam’s activity against strains of *Acinetobacter baumannii* overexpressing *β*-lactamases [73]. Table 2 depicts the newer and next-generation *β*-lactamase inhibitors and their therapeutic combinations.

#### 5.3.6. Cefiderocol

Bacteria, especially Gram-negative *Enterobacterales*, need iron to catalyze various redox pathways, which is fundamental for bacterial growth and survival [74]. Cefiderocol is a fifth-generation cephalosporin that comprises a cephalosporin moiety and a siderophore. The siderophore translates into a site binding to iron, thus utilizing iron transporters to enter into the bacterial cell [75]. By this mechanism, cefiderocol occupies the bacterial cell’s iron-transport system to easily enter the bacterial cell, thus achieving high periplasmic concentrations and its anti-bacterial potential [75]. Furthermore, cefiderocol has a high affinity for penicillin binding proteins 3 (PBP3) while it can resist hydrolysis by various *β*-lactamases [76]. Moreover, cefiderocol shows extended in vitro and in vivo activity against carbapenem-resistant Gram-negative bacteria, with a minimum inhibitory concentration (MIC) lower than 4 mg/L for most *Enterobacterales*, *P. aeruginosa* and *A. baumannii* strains [77]. However, MICs of >8 mg/L have already been reported due to the production of *β*-lactamases and mutations of the iron transport genes, such as *pirA* and *cirA,* which results in resistance to cefiderocol [78]. As cefiderocol has been one of our last resorts regarding carbapenem-resistance *Enterobacterales*, its use must be prudent. It should be noted that it was approved by the Food and Drug Administration (FDA) to treat nosocomial pneumonia and complicated urinary tract infections (cUTIs) in 2019 [77,78].

#### 5.3.7. Eravacycline

Eravacycline is a synthetic fluorocycline antibiotic that is structurally similar to tigecycline with only two modifications in the tetracycline ring. It seems to be a promising agent in vitro against *Acinetobacter baumannii.* However, clinical studies regarding its therapeutic potential against CRAB are still lacking [47].

Overall, despite the lack of development of antimicrobial agents until recently, there is now a great deal of ongoing research. In particular, in the field of newer BLIs, it seems likely that we have truly enriched our armamentarium against multi-drug resistant *Enterobacterales*, carbapenem-resistant *Pseudomonas aeruginosa* and CRAB.

## 6. Pros and Cons of the Study of the Lung Microbiome

The study of the lung microbiome offers us the opportunity to determine the strains and their drug-resistance profiles. The advent of sophisticated molecular techniques, especially whole-genome next-generation sequencing, may provide us with information regarding the microorganisms related to the pathogenesis of aspiration pneumonia as well as their resistance mechanisms, in an accurate and timely fashion. As mentioned above, these modern molecular-based techniques could help us to distinguish between aspiration pneumonia and chemical pneumonitis from aspiration, thus resulting in containing any unnecessary use of antibiotics. However, these molecular-based techniques are rather expensive and further large-scale studies are needed to determine their cost-effectiveness in the field of diagnosis. Regarding aspiration pneumonia, as in other cases of lower respiratory tract infections, researchers have documented a decrease in the diversity of the lung microbiome prior to infection, which may serve as a marker of pneumonia [8]. Therefore, under special circumstances, as previously mentioned, molecular-based techniques provide the opportunity to shed light upon research issues that would otherwise be left unresolved.

## 7. Conclusions

The lung microbiome in healthy adults has been shown to be a de facto phenomenon. In aspiration pneumonia, the lung microbiome is characterized by the predominance of one or a few Gram-negative bacteria. The advent of modern molecular-based techniques, especially mNGS with the use of nanopore sequencing system, is particularly useful with regard to the diagnosis and therapeutics of aspiration pneumonia. Metagenomic NGS with the use of a nanopore sequencing system can identify the causative agent(s) as well as the antimicrobial resistance genes within 6 h. This early recognition of the etiological agent and antimicrobial resistance genes may be life-saving and helpful in containing the global spread of antimicrobial resistance. For all of the above-mentioned reasons, the pros seem to outweigh the cons of these sophisticated techniques, but further large-scale studies are mandatory in order to determine the cost-effectiveness of mNGS. Nevertheless, under special circumstances and in well-equipped and organized laboratories, mNGS may shed light on unanswered scientific questions.

## Figures and Tables

**Figure 1 antibiotics-12-00140-f001:**
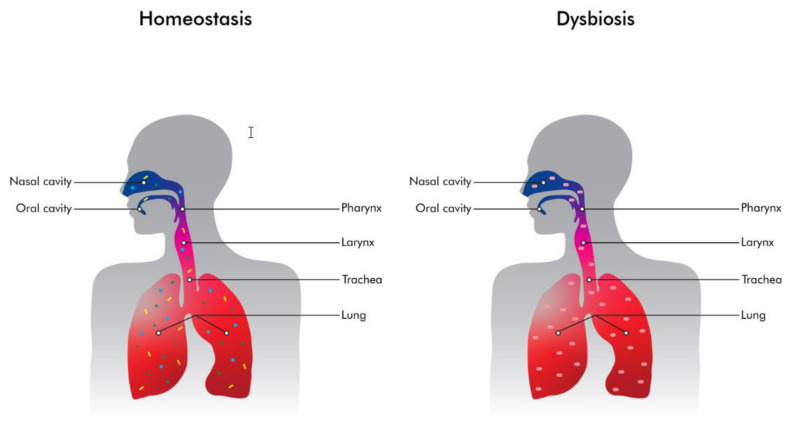
In homeostasis, the lung microbiome is characterized by increased diversity and abundance. On the contrary, when dysbiosis develops, the lung microbiome has reduced diversity and there are a very few or even one predominant species.

**Figure 2 antibiotics-12-00140-f002:**
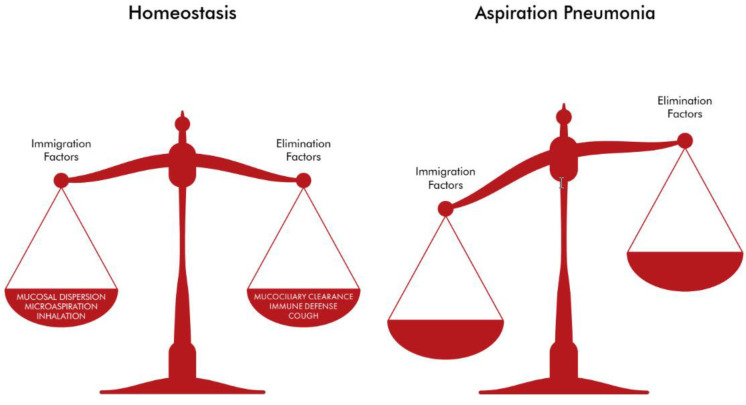
In aspiration pneumonia, there is imbalance between the immigration and elimination factors.

**Table 1 antibiotics-12-00140-t001:** List of risk factors for macroaspiration and development of aspiration pneumonia.

Neurological Causes	Gastrointestinal Causes	Pulmonary Causes
Ischemic Stroke or Intracerebral Hemorrhage	Gastrointestinal Reflux	Chronic Obstructive Pulmonary Disease
Neurogenerative diseases	Oropharyngeal Dysphagia	Mechanical Ventilation Extubation
Parkinsonism	Esophangeal or Gastric Cancer	
Dementia	Esophangeal or Gastric Strictures	
Seizures	Gastrointestinal Motility Disorders	
Multiple Sclerosis	Feeding Tube	
Medications (sleeping pills, antipsychotics, etc.)	Other causes of vomiting (e.g., Cholecystitis, pancreatitis, etc.)	
General Anesthesia		
Alcohol Consumption		
Cardiac Arrest		

**Table 2 antibiotics-12-00140-t002:** List of newer *β*-lactamase Inhibitors and next-generation *β*-lactamase Inhibitors.

Newer *β*-Lactamase Inhibitors	Activity	Therapeutic Combinations
Avibactam	Activity against serine *β*-lactamases, such as class A, class C and some class D BLIs. Not active against *Acinetobacter baumannii*.	Ceftazidime–avibactam Aztreonam–avibactam
Vaborbactam	Activity against some serine *β*-lactamases, such as class A and class C BLIs.	Meropenem–Vaborbactam
Relebactam	Activity against serine *β*-lactamases, such as class A and class C BLIs. No activity against CRAB.	Imipenem–Cilastatin–Relebactam
**Next-generation *β*-lactamase Inhibitors**		
Enmetazobactam	Activity against serine *β*-lactamases, such as class A, class C and class D BLIs.	Cefepime–Enmetazobactam
Zidebactam	Activity against serine *β*-lactamases, such as class A, class C and some MBLs.	Cefepime–Zidebactam
Taniborbactam	Activity against serine-*β*-lactamases (KPC, OXA-48) and MBLs.	Cefepime–Taniborbactam
Nacubactam	Activity against serine *β*-lactamases, such as class A, class C and some class D BLIs	Meropenem–Nacubactam Aztreonam–Nacubactam Cefepime–Nacubactam
Durlobactam	Activity against CRAB	Sulbactam–Durlobactam

Abbreviations: BLIs: *β*-lactamase Inhibitors; CRAB: Carbapenem Resistant *Acinetobacter baumannii*; MBLs: Metallo-*β*-lactamases.

## Data Availability

Not applicable.

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
