# Peer review of "The Role of the Respiratory Microbiome in the Pathogenesis of Aspiration Pneumonia: Implications for Diagnosis and Potential Therapeutic Choices"

_antibiotics, 2023, doi:10.3390/antibiotics12010140_

Round 1

Reviewer 1 Report

1. A brief summary

            The aim of the paper entitled The role of the respiratory microbiome in the pathogenesis of aspiration pneumonia: implications for diagnosis and potential therapeutic choicesis appropriate for consideration of publication in the Antibiotic. This highlighted outcome could be used to positively improve the antimicrobial stewardship for aspiration pneumonia.

2. General concept comments

2.1   The manuscript is moderately structured and written. It has a crucial clinical message and ought to pique the readers' interest. The citing recent references also published within the last 5 years is 65.28% (47/72 publications).

2.2   The details of the article are not completely summarized in the abstract.

2.3   The objective of article cannot be reached through the introduction.

2.4   Your introduction needs more detail. Please add respective literature references on the respiratory infection and pneumonia.

2.5   The sub-topics are not consistent.

Example:

- Are the lungs sterile? = Interrogative sentence

- Diagnosing Aspiration Pneumonia is mainly clinical = Affirmative sentence

- The role of non-cultural methods in describing the lung microbiome = Phrase

2.6   Please correctly compose the figure captions (figure 1-2) according to the academic article style.

2.7   Please offer more details about shotgun metagenomic sequencing for antimicrobial-resistant bacteria regarding to aspiration pneumonia.

2.8   I advise that you improve the description at antimicrobial resistance and responsible use of antimicrobial agents to provide more justification for your evidence and more suggestion for prevent and control of antimicrobial resistance.

2.9   The conclusion section of this review paper is quite vague. Some conclusions should state the primary point of the obtained major content and make recommendations for clinical practice.

2.10 This research's conclusion section requires a suggestion for further research.

2.11 Please double-check the lowercase and uppercase letter usage throughout the article.

Author Response

Dear Sirs,

We would like to thank the Reviewers for their valuable suggestions, which were very useful to improve further the quality of our review.

We have responded to all comments from Reviewers (#1, #2, #3 and #4) and Editors as well as the technical comments. Please find the responses to all comments in a point-by-point fashion below.

Hoping that this letter will meet your kind consideration, we thank you in advance.

Reviewer 1

1. A brief summary

     The aim of the paper entitled “The role of the respiratory microbiome in the pathogenesis of aspiration pneumoniaimplications for diagnosis and potential therapeutic choicesis appropriate for consideration of publication in the Antibiotic. This highlighted outcome could be used to positively improve the antimicrobial stewardship for aspiration pneumonia.

2. General concept comments

    1. The manuscript is moderately structured and written. It has a crucial clinical message and ought to pique the readers' interest. The citing recent references also published within the last 5 years is 65.28% (47/72 publications).

      Author's reply: The text has been edited by a senior author and changes have been made throughout the article in red color.

    1. The details of the article are not completely summarized in the abstract.

      Author's reply: The abstract has been changed according to the Reviewer's suggestions. Abstract: Although the lungs were considered to be sterile until recently, the advent of molecular biology techniques, such as polymerase chain reaction, 16 S rRNA sequencing and metagenomics has led to our expanding knowledge of the lung microbiome. These methods may be particularly useful for the identification of the causative agent(s) in cases of aspiration pneumonia, where there is usually prior administration of antibiotics. As the empirical treatment of aspiration pneumonia is common practice, the usual administration of broad spectrum antibiotics may result in negative cultures from specimens taken from the respiratory tract. Therefore, in such cases, polymerase chain reaction or metagenomics next generation sequencing may be life-saving. Moreover, these modern molecular methods may help for antimicrobial stewardship purposes. Based upon factors, such as age, altered mental consciousness and recent hospitalization, there is a shift towards predominance of aerobes, especially gram negative bacteria over anaerobes in aspiration pneumonia. Thus, the therapeutic choices should be expanded to cover multi-drug resistant gram negative bacteria in selected cases of aspiration pneumonia.

    1. The objective of article cannot be reached through the introduction.

      Author's reply: We have added a paragraph regarding aspiration pneumonia in the Introduction Section. In this paragraph, we have pointed out the difficulties in differentiating aspiration pneumonia from chemichal pneumonitis as well as the significance of non-cultural methods in this differential diagnosis. Aspiration pneumonia refers to the development of lung infection due to specific microorganism(s), which takes place after the aspiration of oropharyngeal or upper gastrointestinal content [8]. It usually locates according to gravity in the superior lower lobe or in the posterior upper lobe segments, if the patient is in the supine position. It should be discriminated from chemical pneumonitis, which refers to the irrigation of the lung parenchyma due to an aspiration event, but not to the inflammation caused by bacteria [8]. It is estimated to account for approximately 15% of cases of community-acquired pneumonia, whereas the estimation rates for nosocomial-acquired cases largely vary due to the heterogeneity of the populations studied [9]. In addition, it may be difficult to diagnose aspiration pneumonia due to the lack of the availability of non-cultural methods in the clinical setting [5].

    1. Your introduction needs more detail. Please add respective literature references on the respiratory infection and pneumonia.

      Author's reply: The Reviewer is absolutely right. We have added the following paragraph together with references 8 and 9 regarding aspiration pneumonia. Aspiration pneumonia refers to the development of lung infection due to specific microorganism(s), which takes place after the aspiration of oropharyngeal or upper gastrointestinal content [8]. It usually locates according to gravity in the superior lower lobe or in the posterior upper lobe segments, if the patient is in the supine position. It should be discriminated from chemical pneumonitis, which refers to the irrigation of the lung parenchyma due to an aspiration event, but not to the inflammation caused by bacteria [8]. It is estimated to account for approximately 15% of cases of community-acquired pneumonia, whereas the estimation rates for nosocomial-acquired cases largely vary due to the heterogeneity of the populations studied [9]. In addition, it may be difficult to diagnose aspiration pneumonia due to the lack of the availability of non-cultural methods in the clinical setting [5].

2.5 The sub-topics are not consistent.

Example:

- Are the lungs sterile?=Interrogative sentence

- Diagnosing Aspiration Pneumonia is mainly clinical=Affirmative sentence

  • The role of non-cultural methods in describing the lung microbiome=Phrase

    Author's reply: We have changed the sub-topics in order to make them as phrases and not interrogation sentences.

    1. Please correctly compose the figure captions (figure 1-2) according to the academic article style.

      Author's reply: We have added the figure captions according to the Reviewer's suggestions seperately in Figure 1 and Figure 2.

    1. Please offer more details about shotgun metagenomic sequencing for antimicrobial-resistant bacteria regarding to aspiration pneumonia.

      Author's reply: For example, Wang et al., with the use of two different mNGS platforms managed to confirm Klebsiella pneumoniae as the causative agent harboring antibiotic resistance genes blaSHV-12, aac(3)-IIa and blaKPC-2. This identification of Klebsiella pneumoniae and its antibiotic resistance genes by mNGS, led to the successful treatment of this immuno-compromized patient with culture-negative pneumonia []. Therefore, under specific circumstances, especially among immuno-compromized patients or in cases of prior use of antibiotics, the use of modern molecular methods seems mandatory in order to establish .....

    1. I advise that you improve the description at antimicrobial resistance and responsible use of antimicrobial agents to provide more justification for your evidence and more suggestion for prevent and control of antimicrobial resistance.

      Author 's reply: We have made several modifications in this regard. For example: Therefore, in order to contain the spread of antibiotic resistance and to be more accurate in decision making regarding the use of antibiotics, it is prudent to send a clinical specimen for identification of the causative agent(s) as well as for susceptibility testing to various antibiotics. However,..

    1. The conclusion section of this review paper is quite vague. Some conclusions should state the primary point of the obtained major content and make recommendations for clinical practice.

      Author 's reply: Metagenomics NGS with the use of nanopore sequencing system may identify the causative agent(s) as well as the antimicrobial resistance genes within 6 hours. This early recognition of the etiological agent and antimicrobial resistance genes may be life-saving and helpful in containing the wide spread of antimicrobial resistance. For all the abivementioned reasons, the pros seems to outweight the cons of these sophisticated techniques, but further large scale studies are mandatory in order to determine the cost-effectiveness of mNGS.

    1. This research's conclusion section requires a suggestion for further research.

      Author's reply: Metagenomics NGS with the use of nanopore sequencing system may identify the causative agent(s) as well as the antimicrobial resistance genes within 6 hours. This early recognition of the etiological agent and antimicrobial resistance genes may be life-saving and helpful in containing the wide spread of antimicrobial resistance. For all the abivementioned reasons, the pros seems to outweight the cons of these sophisticated techniques, but further large scale studies are mandatory in order to determine the cost-effectiveness of mNGS.

2.11 Please double-check the lowercase and uppercase letter usage throughout the article.

Author's reply: We have re-checked the lowercase and uppercase letter usage throughout the manuscript.

Reviewer 2 Report

The article entitled” The role of the respiratory microbiome in the pathogenesis of aspiration pneumonia: implications for diagnosis and potential therapeutic choices” approaches a hot topic in pulmonology, the pulmonary microbiome.

Overall, it is an interesting paper however a present my concerns as follows.

1. In the title you mentioned implications for diagnosis yet there is little information regarding this issue, mostly just enumeration of the techniques used. It would be also useful to present based on the literature review that you performed a hierarchical utility for non-culturable methods. When and where are those useful.

2. Try to develop a part dedicated to culturable methods since most labs still used this method and also tackle the issue of diagnosis of aspiration pneumonia based on culturable techniques.

3. Table 1 is missing in the text.

4. Please include a section for lung microbiome in cystic fibrosis since this disease is known for the colonization produced by several species and perhaps changes in microbiome.

5. Figures 1 and 2 are nice however, they provide little information. Perhaps add another one that summarize the findings you presented for the newer and next generation beta lactamases since that would be really useful.

6. PCR has some limitations like the inability to distinguish between a colonization and infection and also the inability to distinguish between a viable microorganism and a non-viable one. Please address this issue in text as well in section 4.2.

7. Please be specific with the findings provided by Xiao et al regarding the CRAB phenotype. Why are they difficult to distinguish?

All in all, an interesting paper with potential. However improvement is needed in the ”diagnosis” section that is approached in the title.

Also it would be good to emphasize the strengths, originality and limitations of the study.

Author Response

Dear Sirs,

We would like to thank the Reviewers for their valuable suggestions, which were very useful to improve further the quality of our review.

We have responded to all comments from Reviewers (#1, #2, #3 and #4) and Editors as well as the technical comments. Please find the responses to all comments in a point-by-point fashion below.

Hoping that this letter will meet your kind consideration, we thank you in advance.

Reviewer 2

The article entitled” The role of the respiratory microbiome in the pathogenesis of aspiration pneumonia: implications for diagnosis and potential therapeutic choices” approaches a hot topic in pulmonology, the pulmonary microbiome.

Overall, it is an interesting paper however a present my concerns as follows.

  1. In the title you mentioned implications for diagnosis yet there is little information regarding this issue, mostly just enumeration of the techniques used. It would be also useful to present based on the literature review that you performed a hierarchical utility for non-culturable methods. When and where are those useful.

    Author's reply: We have re-written the diagnosis section in order to improve our manuscript according to the Reviewer's suggestions. For example: Therefore, the advent of non-cultural molecular techniques for studying the lung microbiome really revolutionized our knowledge of the diversity of the lung microbiome in healthy adults. PCR, 16 S rRNA gene sequencing and especially whole generation sequencing and metagenomics, which could provide us with information to the species level, all have contributed largely to our defining the composition of the lung microbiome in healthy individuals [24-25]. Shotgun metagenomic sequencing give us the opportunity to sequence all genetic material within a sample, thereby, allowing for a holistic and deeper taxonomic characterization of the lung microbiome to the species or even strains level [24-25].

  1. Try to develop a part dedicated to culturable methods since most labs still used this method and also tackle the issue of diagnosis of aspiration pneumonia based on culturable techniques.

    Author's reply: Apart from gram staining, the clinical specimen is being cultured on an agar plate and then left for an overnight aerobic incubation at 37C. After bacterial culture, and based upon the various biochemical features of the bacteria, identification to the species level is performed, usually with the use of API Systems. Nowadays, identification to the species level may be achieved for even more complex microorganisms, with the use of the more sophisticated MALDI-TOF MS (Matrix-Assisted Laser Desorption and Ionization Time of Flight Mass Spectrometry) [32]. In addition, antibiotic susceptibility testing mainly with disk diffusion technique is most usually performed.

  1. Table 1 is missing in the text.

    Author's reply: Risk factors related to aspiration pneumonia are listed in Table 1.

  1. Please include a section for lung microbiome in cystic fibrosis since this disease is known for the colonization produced by several species and perhaps changes in microbiome.

    Author's reply: The diversity of the lung microbiome increases with age, when compared to the children's lung microbiome, as is the case of the gut microbiome. However, for example, in the case of cystic fibrosis, Linnane et al., have demonstrated a decrease in the diversity of the lung microbiome with age among patients with cystic fibrosis. In addition, they have shown that Pseudomonas and Staphylococcus were more abundant with age. On the contrary, Streptococcus, Porphyromonas and Veillonella were lesss frequently encountered with age among patients with cystic fibrosis [23]

  1. Figures 1 and 2 are nice however, they provide little information. Perhaps add another one that summarize the findings you presented for the newer and next generation beta lactamases since that would be really useful.

    Author's reply: The Reviewer is right. Therefore, we have added Table 2, which summarizes newer and next generation beta lactamases.

  1. PCR has some limitations like the inability to distinguish between a colonization and infection and also the inability to distinguish between a viable microorganism and a non-viable one. Please address this issue in text as well in section 4.2.

    Author's reply: PCR's limitations have been added in the text as follows: However, PCR has some limitations, such as its inability to discriminate between viable and non-viable microorganisms, while it may not be able to distinguish between a colonization and an infection.

  2. Please be specific with the findings provided by Xiao et al regarding the CRAB phenotype. Why are they difficult to distinguish?Author's reply: In particular, as Acinetobacter baumannii is a ubiquitous microorganism, it may be frequently isolated from the respiratory tract in patients with tracheal intubation. However, this does not necessary translates into the existence of VAP due to Acinetobacter baumannii, but may merely reflect just a colonization. On the other hand, an accurate diagnosis of CRAB pneumonia is mandatory, as it is associated with substantial mortality. However, PCR has some limitations, such as its inability to discriminate between viable and non-viable microorganisms, while it may not be able to distinguish between a colonization and an infection. Therefore, to distinguish between CRAB colonization and infection, and thereby to reduce any unnecessary use of antibiotics, Xiao et al., used multi-omics analysis with the performance of 16 S rRNA amplicon analysis, metagenomics sequencing and whole genome sequencing (WGS) [38]. These techniques with the advent of specific platforms and bioinformatics are very helpful in providing us with information regarding CRAB as a colonizer or a pathogen in respiratory tract infections. They concluded that patients with CRAB pneumonia were characterized by decreased diversity in the lung microbiome, increased relative abundance of Acinetobacter as well as increased existence of virulence factors [38].

All in all, an interesting paper with potential. However improvement is needed in the ”diagnosis” section that is approached in the title.

Author's reply: We have made several modifications in the Diagnosis section according to the Reviewers' suggestions. For example: Apart from gram staining, the clinical specimen is being cultured on an agar plate and then left for an overnight aerobic incubation at 37C. After bacterial culture, and based upon the various biochemical features of the bacteria, identification to the species level is performed, usually with the use of API Systems. Nowadays, identification to the species level may be achieved for even more complex microorganisms, with the use of the more sophisticated MALDI-TOF MS (Matrix-Assisted Laser Desorption and Ionization Time of Flight Mass Spectrometry) [32]. In addition, antibiotic susceptibility testing mainly with disk diffusion technique is most usually performed.

However, PCR has some limitations, such as its inability to discriminate between viable and non-viable microorganisms, while it may not be able to distinguish between a colonization and an infection.

In particular, as Acinetobacter baumannii is a ubiquitous microorganism, it may be frequently isolated from the respiratory tract in patients with tracheal intubation. However, this does not necessary translates into the existence of VAP due to Acinetobacter baumannii, but may merely reflect just a colonization. On the other hand, an accurate diagnosis of CRAB pneumonia is mandatory, as it is associated with substantial mortality. However, PCR has some limitations, such as its inability to discriminate between viable and non-viable microorganisms, while it may not be able to distinguish between a colonization and an infection. Therefore, to distinguish between CRAB colonization and infection, and thereby to reduce any unnecessary use of antibiotics, Xiao et al., used multi-omics analysis with the performance of 16 S rRNA amplicon analysis, metagenomics sequencing and whole genome sequencing (WGS) [38]. These techniques with the advent of specific platforms and bioinformatics are very helpful in providing us with information regarding CRAB as a colonizer or a pathogen in respiratory tract infections. They concluded that patients with CRAB pneumonia were characterized by decreased diversity in the lung microbiome, increased relative abundance of Acinetobacter as well as increased existence of virulence factors [38].

Also it would be good to emphasize the strengths, originality and limitations of the study.

Author's reply: In the Pros and Cons section, just before the Conclusion section, we have stated the major advantages as well as the disadvantages of these sophsticated techniques. As this is a review manuscript and not a research one, we have not added any strengths or limitations of this manuscript. However, regarding the originality of this review article, there are very few references regarding aspiration pneumonia and the lung microbiome in databases, such as Google Scholar or Pubmed.

Reviewer 3 Report

Thank you for providing an opportunity to review this manuscript. This study could be help in understanding the role of lung microbiome in conditions like aspiration pneumonia. 

Authors are requested to please avail the services of senior co-author for English language and corrections. 

Author should consider to add little bit about aspiration pneumonia in the introduction section. That would help to build flow and connectivity wit the study findings. 

Please delete repeating word 'among' before references 11-14 in subheading 2.1.

Please correct the word environmental under heading 'The Lung Microbiome in Aspiration Pneumonia'. 

Author Response

Dear Sirs,

We would like to thank the Reviewers for their valuable suggestions, which were very useful to improve further the quality of our review.

We have responded to all comments from Reviewers (#1, #2, #3 and #4) and Editors as well as the technical comments. Please find the responses to all comments in a point-by-point fashion below.

Hoping that this letter will meet your kind consideration, we thank you in advance.

Reviewer 3

Thank you for providing an opportunity to review this manuscript. This study could be help in understanding the role of lung microbiome in conditions like aspiration pneumonia.

Authors are requested to please avail the services of senior co-author for English language and corrections.

Author's reply: Senior author's advise was requested throughout the text.

Author should consider to add little bit about aspiration pneumonia in the introduction section. That would help to build flow and connectivity wit the study findings.

Author's reply: The Reviewers suggestion was followed accordingly: Aspiration pneumonia refers to the development of lung infection due to specific microorganism(s), which takes place after the aspiration of oropharyngeal or upper gastrointestinal content [8]. It usually locates according to gravity in the superior lower lobe or in the posterior upper lobe segments, if the patient is in the supine position. It should be discriminated from chemical pneumonitis, which refers to the irrigation of the lung parenchyma due to an aspiration event, but not to the inflammation caused by bacteria [8]. It is estimated to account for approximately 15% of cases of community-acquired pneumonia, whereas the estimation rates for nosocomial-acquired cases largely vary due to the heterogeneity of the populations studied [9]. In addition, it may be difficult to diagnose aspiration pneumonia due to the lack of the availability of non-cultural methods in the clinical setting [5].

Please delete repeating word 'among' before references 11-14 in subheading 2.1.

Author's reply: The Reviewer is absolutely right. We have omitted the word “among”, which was written twice before.

Please correct the word environmental under heading 'The Lung Microbiome in Aspiration Pneumonia'.

Author's reply: The word “environmental” has been corrected, as mentioned from the Reviewer.

Reviewer 4 Report

1.      “Hilty et all” write as “Hilty et al.,”

2.      Please add a table or figure showing the normal lungs microbiota

3.      Antibiotic resistance is another important issue, which type of resistance upto yet is mainly reported from lungs microbiota; please write the mechanisms or antibiotic-resistant genes names

4.      In microbiota, is there’s any fungal species reported as normal flora of lungs, for example as candida albican is normal flora and an opportunistic pathogen of the skin, Are any such cases for lungs?

5.      Darie et al replaced by Darie et al., It is also ok if you do not write the author’s names in the text, in the references is also enough

6.      The recommendation of any antibiotics prior to identifying organism and their resistant level is not appropriate in any way. YOU SHOULD HAVE TO MENTION THAT FOR ANY KIND OF ANTIBIOTICS ADMANISTRASTION, first, it needs to identify the bacterial type and its susceptibility profile and then need to recommend antibiotics on the basis of the bacterial profile. After that, you can add your empirical treatment guidelines.

Author Response

Dear Sirs,

We would like to thank the Reviewers for their valuable suggestions, which were very useful to improve further the quality of our review.

We have responded to all comments from Reviewers (#1, #2, #3 and #4) and Editors as well as the technical comments. Please find the responses to all comments in a point-by-point fashion below.

Hoping that this letter will meet your kind consideration, we thank you in advance.

Reviewer 4

Hilty et all” write as “Hilty et al.,”

Authors' reply: The first author's name has been changed accordingly throughout the text. For example: Hilty et al.,

  1. Please add a table or figure showing the normal lungs microbiota.

    Author reply: As normal lung microbiota differs between individuals, just like the gut microbiota differ from one person to the other, we have not added a table/figure with the normal flora, as in our opinion, this addition would have been highly speculative and perhaps inaccurate or very general.

  1. Antibiotic resistance is another important issue, which type of resistance upto yet is mainly reported from lungs microbiota; please write the mechanisms or antibiotic-resistant genes names

    Author's reply: We have added also a Table 2 regarding beta-lactamases and their therapeutic combinations along with their spectrum of activity. In addition, we have added the following: For example, Wang et al., with the use of two different mNGS platforms managed to confirm Klebsiella pneumoniae as the causative agent harboring antibiotic resistance genes blaSHV-12, aac(3)-IIa and blaKPC-2. This identification of Klebsiella pneumoniae and its antibiotic resistance genes by mNGS, led to the successful treatment of this immuno-compromized patient with culture-negative pneumonia []. Therefore, under specific circumstances, especially among immune-compromised patients or in cases of prior use of antibiotics,...

  1. In microbiota, is there’s any fungal species reported as normal flora of lungs, for example as candida albican is normal flora and an opportunistic pathogen of the skin, Are any such cases for lungs?

Author's reply: This is an intriguing question, indeed! However, we did not find any references regarding Candida species or other fungi in the normal lung microbiome. However, there are a few references in lung diseases, such as lung cancer and COPD.

  1. Darie et al replaced by Darie et al., It is also ok if you do not write the author’s names in the text, in the references is also enough

    Authors' reply: Names of first authors have been changed accordingly throughout the text. For Example: Darie et al.,

6. The recommendation of any antibiotics prior to identifying organism and their resistant level is not appropriate in any way. YOU SHOULD HAVE TO MENTION THAT FOR ANY KIND OF ANTIBIOTICS ADMANISTRASTION, first, it needs to identify the bacterial type and its susceptibility profile and then need to recommend antibiotics on the basis of the bacterial profile. After that, you can add your empirical treatment guidelines.

Author's reply: The Reviewer is right. Therefore, in order to contain the spread of antibiotic resistance and to be more accurate in decision making regarding the use of antibiotics, it is prudent to send a clinical specimen for identification of the causative agent(s) as well as for susceptibility testing to various antibiotics. However,...

Round 2

Reviewer 1 Report

Thank you for your thoughtful reply. The manuscript modified as per the comments and were satisfactory. Consequently, this manuscript is of adequate quality to be published.

Reviewer 2 Report

Accept in curent form.